# Acute Limb Ischemia after Intake of the Phenylethylamine Derivate NBOMe

**DOI:** 10.3390/ijerph16245071

**Published:** 2019-12-12

**Authors:** Patricia P. Wadowski, Georgiana-Aura Giurgea, Oliver Schlager, Anton Luf, Thomas Gremmel, Eva-Luise Hobl, Sylvia Unterhumer, Henriette Löffler-Stastka, Renate Koppensteiner

**Affiliations:** 1Division of Angiology, Department of Internal Medicine II, Medical University of Vienna, A-1090 Vienna, Austria; patricia.wadowski@meduniwien.ac.at (P.P.W.); georgiana-aura.giurgea@meduniwien.ac.at (G.-A.G.); oliver.schlager@meduniwien.ac.at (O.S.); thomas.gremmel@meduniwien.ac.at (T.G.); 2Department of Laboratory Medicine, Medical University of Vienna, A-1090 Vienna, Austria; anton.luf@meduniwien.ac.at; 3Department of Clinical Pharmacology, Medical University of Vienna, A-1090 Vienna, Austria; eva-luise.hobl@akhwien.at; 4Department of Biomedical Imaging and Image-guided Therapy, Medical University of Vienna, A-1090 Vienna, Austria; sylvia.unterhumer@akhwien.at; 5Department of Psychoanalysis and Psychotherapy, Medical University of Vienna, A-1090 Vienna, Austria; henriette.loeffler-stastka@meduniwien.ac.at

**Keywords:** upper limb pain, limb ischemia, vasospasm, phenylethylamine derivates

## Abstract

*Objective*: N-(2-methoxy) benzyl-phenethylamine (NBOMe) derivatives have a high affinity to the serotonin receptor 2A and emerged as new psychedelic agents. We report the case of a 30-year-old man admitted to the hospital because of acute ischemia of the left arm with clinical symptoms of pallor, pulselessness, paresthesia, and a motoric deficit. The patient had a history of schizophrenia and drug abuse and disclosed during the hospital stay the sublingual intake of a substance bought as 25I-NBOMe the night before the ischemic event. *Methods*: Routine clinical diagnostics including among others color-coded duplex sonography and computed tomography angiography (CTA) were performed. The remainder of the drugs was analyzed using high performance liquid chromatography. *Results*: Initial color-coded duplex sonography of the upper left limb showed pathological flow profiles of the axillary, brachial, ulnar, and radial artery with a reduced diameter of the ulnar (0.9 mm) and radial (1.1 mm) artery. In consequence, peripheral vasospasm, distal arterial thrombosis, or arterial embolization was anticipated. As therapeutic measures, the patient immediately received intravenous systemic vasodilators (alprostadil) and therapeutic anticoagulation with low molecular weight heparin. Instant symptom improvement was observed within the first day after therapy initiation. The subsequently performed CTA of the heart and left arm showed no signs of thrombotic material. Treatment was continued for five days and the patient was released thereafter having completely normalized perfusion in his left arm. Outpatient treatment was continued with calcium-channel blockers, as the patient had also displayed arterial hypertension. Drug analysis retrieved a composition of the isomers 25I-NBOMe, 25C-NBOMe, and 25H-NBOMe as well as traces of pentylon. Conclusion: NBOMe ingestion implicates the risk of peripheral vasospasms with severe, limb-threatening ischemia.

## 1. Introduction

NBOMes are N-(2-methoxy) benzyl derivates of the 2C family of hallucinogens and emerged as new psychoactive substances. [1,2] They were initially synthetized for research utilization as agonists of the serotonin receptor 2A, exhibiting high affinity through the N-benzyl moiety. [2,3] The hallucinogenic effects are evoked through structure similarity to mescaline, further methoxy substitutions at the positions 2 and 5 of the phenylethylamine structure, and possible halogen substitution at position 4 with bromide, iodine or chlorine [4]. Because of the high affinity to the serotonin receptor, NBOMes exhibit a high risk of overdosing and doses as low as 50 µg may already lead to psychedelic effects [4,5]. In the literature, cases with adverse effects after NBOMe abuse are reported since 2013 with a considerable number of intensive care admissions and fatal poisonings [4,6,7]. The most common signs of intoxication are tachycardia and agitation (each 85%), hypertension (65%), dilated pupils (55%), seizures, delirium, and hallucination (each 40%) [4].

### Case Presentation

A 30-year-old man presented with pallor and paresthesia of the left arm at the Division of Angiology of the Medical University of Vienna. Clinical examination revealed pulselessness, pain, hypaesthesia, and a slight motoric deficit of the left upper extremity. The patient was suspected of acute vasospasm, arterial thrombosis, or arterial embolization, and therefore hospitalized immediately. He reported sublingual ingestion of a new illicit substance purchased online. At the time of admission the patient did not remember the substance name. Moreover, he reported the intake of methylone, ethylone, lysergic acid diethylamide (LSD), and ecstasy as well as cigarette smoking. Because of schizophrenia, the patient was on quetiapine and amisulpride therapy.

## 2. Methods

### 2.1. Routine Medical and Clinical Diagnostics

Physical examination of the patient was performed at arrival. Consequently, routine non- invasive vascular tests were applied including pneumatic segmental pulse and optical oscillography as well as color-coded duplex sonography and 24 h blood pressure monitoring. Furthermore, transthoracal echocardiography and a computed tomography angiography (CTA; using iomeprol as contrast agent) were performed. Routine laboratory tests including toxicological screening were performed at the Department of Laboratory Medicine of the Medical University of Vienna on the cobas 6000 (c501) analyzer, Roche Diagnostics GmbH, Munich, Germany.

### 2.2. NBOMe Sample Analysis

Samples of the NBOMe substance were analyzed on a LC-Packing Ultimate dual high performance liquid chromatographic (HPLC) system (Dionex, Amsterdam, Netherlands) equipped with four Agilent 1200 photodiode array detectors (Agilent Technologies, Santa Clara, CA, USA). The separation system was coupled in-line with a Thermo LTQ Velos linear ion trap mass spectrometer (Thermo Electron Corporation, San Jose, CA, USA) equipped with an electrospray ionization (ESI) probe. All separations were performed on 2.1 × 150 mm columns packed with Kinetex F5 2.6 μm (Phenomenex, Torrance, CA, USA). Two microliters of sample solution was injected into the HPLC system. After separation, compounds were detected simultaneously by the photodiode array detection (PDA) and the mass spectrometer, if eligible. The operation of the liquid chromatography-mass spectrometry (LC–MS) and chromatographic analysis was controlled by a Chromeleon 6.8 (SR 14) chromatography software (Dionex, Sunnyvale, CA, USA Germany). For the identification of compounds, retention time, UV spectra, and mass spectra were obtained and compared to those of the substances previously measured. Quantitation data were obtained by integration of the UV-Trace at a wavelength of 215 and 254 nm using an internal standard quantitation method [8].

## 3. Results

### 3.1. Routine Medical and Clinical Diagnostics

Pneumatic segmental pulse oscillography showed a flatline oscillogram when positioned at the left wrist (Figure 1a). A flatline optical oscillogram was also obtained of all left fingers (Figure 1b). Immediate color-coded duplex sonography of the upper left limb was performed showing pulsatile flow in the subclavian artery, but pathological flow profiles suggestive of distal flow obstacles in the axillary, brachial, radial, and ulnar arteries with reduced diameters of the radial (1.1 mm) and ulnar (0.9 mm) artery (Figure 2). Laboratory serum analyses showed elevated creatinine kinase (253 U/L), myoglobin (126 ng/mL), and lactate dehydrogenase levels (298 U/L).

Further documents of the patient were retrieved from the database of the General hospital of Vienna and another Viennese hospital and revealed cocaine intake the month before. A urinary drug screening was performed and showed positive results for amphetamine, buprenorphine, cotinine, and benzodiazepine. Cocaine was not present in the urinary drug screening. During the hospital stay the patient admitted sublingual ingestion of the substance NBOMe. The patient provided the remainder of the drug samples of NBOMe for laboratory analyses three weeks later. NBOMe sample analysis is presented in Section 3.2.

Two days after admission a CTA of the aorta and left upper extremity was performed and showed normal diameters of the arteries of the left arm with normal perfusion; no thrombi or plaques as potential sources of distal embolization could be detected (Figure 3). Furthermore a transthoracic echocardiography was performed, which was normal. The control color-coded duplex sonography at the end of hospitalization confirmed the findings of the CTA and exhibited regular flow profiles in the arteries of the upper left extremity. A control optical oscillogram showed normal oscillations at the fingers of both hands (Figure 4).

Because of the newly diagnosed arterial hypertension and anamnestic acute renal failure after ecstasy consumption, the patient underwent renal ultrasound. The kidneys were of regular form and size with no stenosis of the renal artery and a regular resistance index (0.65 for the left and 0.66 for the right renal segmental arteries).

### 3.2. NBOMe Sample Analysis

In the four samples sold as “1g 25I-NBOMe” HPLC retrieved the following results:Sample 1: 25I-NBOMe—923 mg/g, 25C-NBOMe—17 mg/g, 25H-NBOMe—16 mg/g and traces of the substance pentylon.Sample 2: 25I-NBOMe—914 mg/g, 25C-NBOMe—20 mg/g, 25H-NBOMe—16 mg/g.Sample 3: 25I-NBOMe—941 mg/g, 25C-NBOMe—23 mg/g, 25H-NBOMe—20 mg/g.Sample 4: 25I-NBOMe—951 mg/g, 25C-NBOMe—22 mg/g, 25H-NBOMe—19 mg/g.

### 3.3. Therapeutic Measures and Outcome

Because of the suspected acute vasospasm, arterial thrombosis (or arterial embolization) the patient received therapeutic low molecular weight heparin and intravenous alprostadil instantly. The patient’s symptoms improved constantly during the hospital stay, the motoric function restored completely within the first day of treatment initiation and the patient reported merely occasional paresthesia in the left forefinger. The patient was discharged after five days in hospital with calcium-channel blockers as medication. One week later, a 24 h blood pressure monitoring was performed in the outpatient clinic, which showed hypertension with a flattened 24 h-blood pressure profile (non-dipper). In the further course, however, he did not appear to the next control visit, which was scheduled four weeks later, and it was also not possible for us to establish a telephone contact to him to evaluate the clinical outcome and to do therapeutic adjustments.

## 4. Discussion

This case report aims at raising the awareness of severe limb-threatening complications of drug abuse, in particular novel substances, which can be easily purchased via the internet. NBOMe isomers have occurred on the illicit drug market as a new psychedelic substance [3]. Effects are observed after doses of 50 µg and overdosing as well as fatal poisoning is extremely facile [5]. The most frequent side effects include tachycardia, agitation, and hypertension [4]. The symptoms of our patient after ingestion of NBOMe isomers resembled clinical presentation of cocaine consumption with peripheral vasospasms and severe, limb-threatening ischemia [9,10]. However, as serotonin receptor pathway agonists, NBOMes as well as LSD may evoke vasoconstriction [11]. Furthermore, the combination of the ingested substances including amphetamines (i.e., ecstasy) might have led to the distinct clinical symptoms, potentiating vasoconstriction [12].

This limits our observation, as the patient’s clinical presentation cannot only be attributed to NBOMe ingestion. We ascribe the symptoms mostly to the substance NBOMe, as it was new in the patient’s anamnesis. Furthermore, traces of cocaine, which is also known to evoke symptoms alike [9,10] were not found in the urinary drug screening.

Other side effects observed in our patient were hypertension and elevated creatinine kinase levels, which were also described previously to be NBOMe associated [4].

The patient was discharged after complete resolution of the vasospasm as demonstrated by duplex sonography and CTA and exclusion of a thromboembolic event. We decided to treat hypertension with a calcium channel antagonist in analogy to medication in patients with (secondary) Raynaud’s phenomenon [13]. After hospital discharge the patient brought the remainder of the drug. The outer packing of the substances, which were originally posted, was only a non-tightly closed bag. Because of the drug affinity to its receptors [3], the inhaled doses or the absorption through the skin might already lead to toxicity and therefore also exhibit danger to the mail carrier.

Nevertheless, an important question remains: the bio-psycho-social long-term treatment approach and health literacy of this patient. As schizophrenia is a severe mental disorder with reduced life expectancy and high mortality because of physical illness, especially cardiovascular disease, the unhealthy lifestyle of this patient should be focused [14].

Individual health promotion is evidence-based practice [15], and included in guidelines [16,17], but individuals’ health literacy is the most important key-factor for modification. Different moderators and mediators for the efficacy of healthy lifestyle interventions are known for patients with severe mental disorders and should be explored further [18].

## 5. Conclusions

Intake of the phenylethylamine derivate NBOMe should be considered as a risk factor for peripheral vasospasm with severe, limb-threatening complications in routine medical practice.

## Figures and Tables

**Figure 1 ijerph-16-05071-f001:**
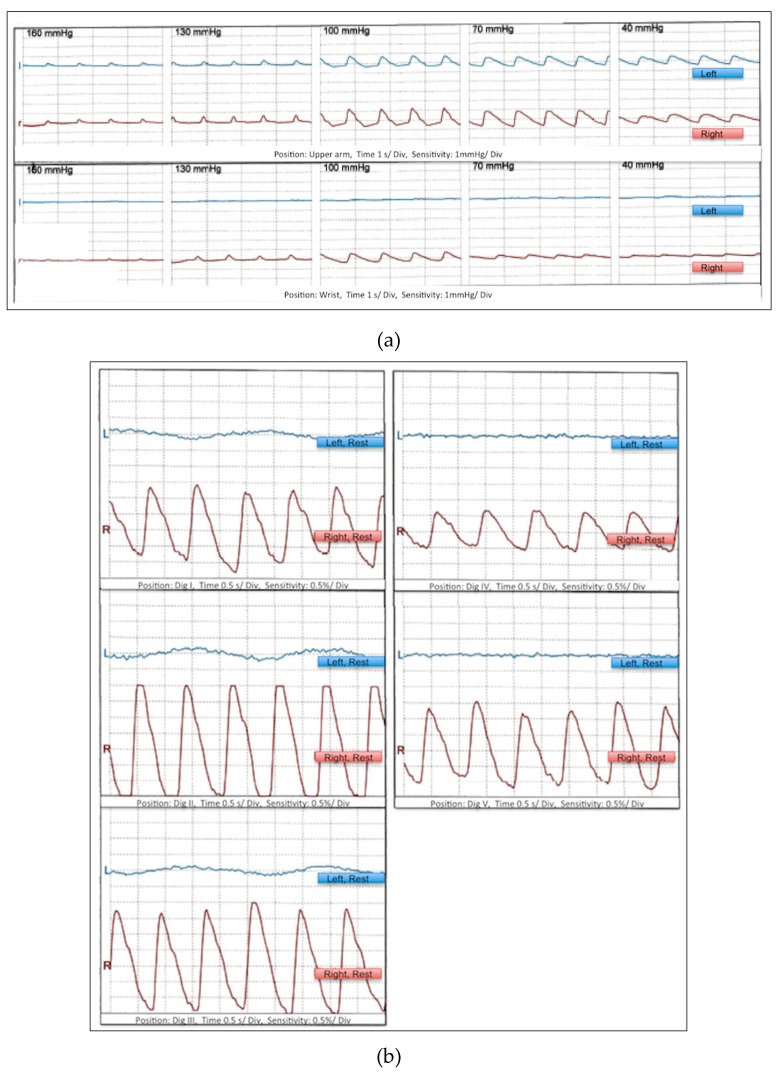
Pneumatic segmental pulse oscillography positioned at the left wrist (**a**) and optical oscillography of all fingers of the left hand (**b**) showed a flatline.

**Figure 2 ijerph-16-05071-f002:**
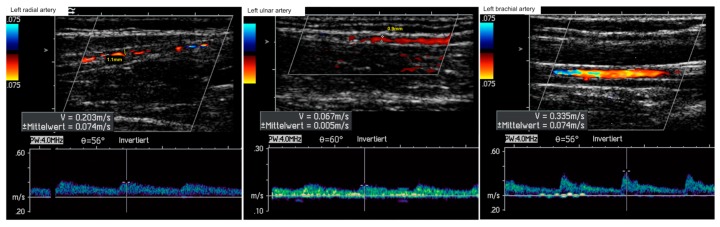
Color-coded duplex sonography with flow profiles of the arteries of the left upper extremity on admission.

**Figure 3 ijerph-16-05071-f003:**
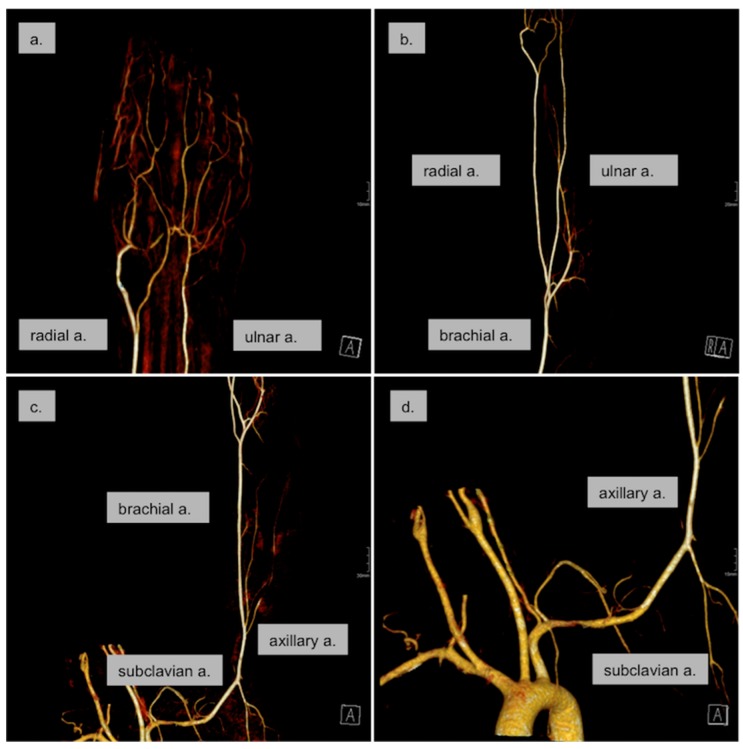
Computed tomography angiography (CTA) two days after initiation of treatment. A complete resolution of the peripheral vasospasm without a distal thrombus was seen (**a**,**b**). There were no sources for a possible arterial embolization in the subclavian, axillary, and brachial artery (**c**) or the aortic arch (**d**).

**Figure 4 ijerph-16-05071-f004:**
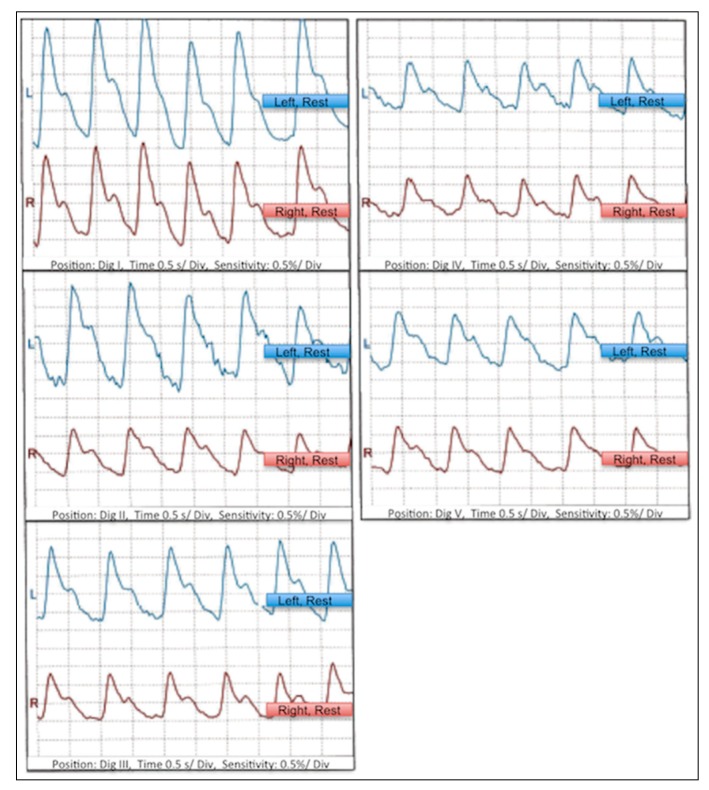
Control optical oscillogram before discharge showing regular oscillations at the fingers of both hands.

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
