# Peer review of "Acute Limb Ischemia after Intake of the Phenylethylamine Derivate NBOMe"

_ijerph, 2019, doi:10.3390/ijerph16245071_

Round 1

Reviewer 1 Report

This reviewer honestly appreciates the work of the authors related to this manuscript, but the following concerns should be addressed:

Please reconsider the title of the manuscript. Think about including basic information about what NBOMes are (phenylethylamine derivates emerging as new psychedelic agents). Do you think it is necessary to mention all the isomers in the title?

The abstract requires for some more structure. I suggest to offer a definition for NBOMe in a first sentence, but in my opinion it is not required to mention the 25I-NBOMe in detail in the Objective. 

Line 23: This reviewer recommend to change " he" to "the patient".

If applicable, please include flow profiles in figure 2.

Line 67/68 is a complete repetition (see line 55/56). Consider to delete.

Please consider to revise the detailed information about figure 3 (as provided in line 91/92).

Line 95 is a repetition again.

I guess section 2.1. Sample analysis is about NBOMe-analysis only, please note.

In the abstract, Methods and Results include information about all diagnostic and therapeutic steps, but in the manuscript, these both sections offer methodological performance and test results of the NBOMe liquid chromatography only. Please take care for consistency about which information to provide in which section. 

Line 130: Rethink the phrase "The substance NBOMe...". (Perhaps use "NBOMe isomers...)

There are some major concers raised about the discussion:

1, There is no conclusion offered, please include. 

2,  Please explain why the authors traced the ischemic event to NBOMe. Do you think it is possible that the vasospasm also resulted from other detected substances or from the combination of detected drugs? Please address this concern more detailed.

3, The last paragraph focusses on schizophrenia and mental disorders. Consider to shorten as this topic is not mentioned at all in other sections of your manuscript. 

Author Response

To the Editor-in-Chief

                                                                                              Vienna, 30 November 2019

Dear Professor Tchounwou,

We would like to submit a revised version of our manuscript, entitled Acute Limb Ischemia after Intake of the Phenylethylamine Derivate NBOMe“ (Manuscript ID: ijerph-631953), for publication in the International Journal of Environmental Research and Public Health.

We would like to thank you and the referees for the review and the valuable comments and suggestions. The manuscript has been amended accordingly. We believe that these changes have resulted in a greatly improved manuscript, which we hope is now suitable for publication in International Journal of Environmental Research and Public Health.

Yours sincerely,

Patricia Wadowski, MD

Point-by-point response to Reviewer 1:

Thank you very much for carefully reading our manuscript and your valuable suggestions, which we have followed.

Open Review

English language and style

( ) Extensive editing of English language and style required
(x) Moderate English changes required
( ) English language and style are fine/minor spell check required
( ) I don't feel qualified to judge about the English language and style

Yes

Can be improved

Must be improved

Not applicable

Does the introduction provide sufficient background and include all relevant references?

(x)

( )

( )

( )

Is the research design appropriate?

(x)

( )

( )

( )

Are the methods adequately described?

( )

(x)

( )

( )

Are the results clearly presented?

( )

(x)

( )

( )

Are the conclusions supported by the results?

( )

( )

(x)

( )

Comments and Suggestions for Authors

This reviewer honestly appreciates the work of the authors related to this manuscript, but the following concerns should be addressed:

Please reconsider the title of the manuscript. Think about including basic information about what NBOMes are (phenylethylamine derivates emerging as new psychedelic agents). Do you think it is necessary to mention all the isomers in the title?

Authors´ response: Thank you for the valuable comment, we have now changed the title to “Acute Limb Ischemia after Intake of the Phenylethylamine Derivate NBOMe”

The abstract requires for some more structure. I suggest to offer a definition for NBOMe in a first sentence, but in my opinion it is not required to mention the 25I-NBOMe in detail in the Objective. 

Authors´ response: As suggested, we have included a definition of NBOMe in the first sentence (page 1, lines 15-16 of the revised manuscript).

NBOMes are phenylethylamine derivates with a high affinity to the serotonin receptor 2A and emerged as new psychedelic agents.

Line 23: This reviewer recommend to change " he" to "the patient".

Authors´ response: We have amended the sentence accordingly (page 1, lines 23-25 of the revised manuscript).

As therapeutic measures, the patient immediately received intravenous systemic vasodilators (alprostadil) and therapeutic anticoagulation with low molecular weight heparin.

If applicable, please include flow profiles in figure 2.

Authors´ response: We have now included flow profiles in Figure 2 (page 4 of the revised manuscript).

Figure 2. Color-coded duplex sonography with flow profiles of the arteries of the left upper extremity on admission.

Line 67/68 is a complete repetition (see line 55/56). Consider to delete.

Authors´ response: We have now deleted this sentence.

Please consider to revise the detailed information about figure 3 (as provided in line 91/92).

Authors´ response: We have now revised the detailed information about figure 3 (page 5, lines 122-125 of the revised manuscript).

Computed tomography angiography (CTA) two days after initiation of treatment. A complete resolution of the peripheral vasospasm without a distal thrombus was seen (a+b). There were no sources for a possible arterial embolization in the subclavian, axillary and brachial artery (c) or the aortic arch (d).

Line 95 is a repetition again.

Authors´ response: We have now deleted the sentence.

I guess section 2.1. Sample analysis is about NBOMe-analysis only, please note.

Authors´ response: Thank you for the comment, we have now restructured the methods and included this information into the section 2.2. This is now mentioned on page 2, lines 68-71 of the revised manuscript.

2.2 NBOMe sample analysis:

Samples of the NBOMe substance were analyzed on a LC-Packing Ultimate dual high performance liquid chromatographic (HPLC) system (Dionex, Netherlands) equipped with four Agilent 1200 photodiode array detectors (Agilent Technologies, Santa Clara).

In the abstract, Methods and Results include information about all diagnostic and therapeutic steps, but in the manuscript, these both sections offer methodological performance and test results of the NBOMe liquid chromatography only. Please take care for consistency about which information to provide in which section. 

Authors´ response: We have now restructured the Introduction, Methods and Results and included a paragraph on the routine medical and clinical diagnostics applied in the Methods and Results, respectively.

Line 130: Rethink the phrase "The substance NBOMe...". (Perhaps use "NBOMe isomers...)

Authors´ response: As suggested, we have amended the phrase (page 7, lines 151-152 of the revised manuscript)

NBOMe isomers have occurred on the illicit drug market as a new psychedelic substance. [1]

There are some major concers raised about the discussion:

1, There is no conclusion offered, please include. 

Authors´ response: Thank you for the important comment, we have now added a conclusion (page 7, lines 182-184 of the revised manuscript).

Conclusion

Intake of the phenylethylamine derivate NBOMe should be considered as a risk factor for peripheral vasospasm with severe, limb-threatening complications in routine medical practice.

2,  Please explain why the authors traced the ischemic event to NBOMe. Do you think it is possible that the vasospasm also resulted from other detected substances or from the combination of detected drugs? Please address this concern more detailed.

Authors´ response: As suggested, we have now elaborated the discussion (page 7, lines 154-163 of the revised manuscript).

The symptoms of our patient after ingestion of NBOMe isomers resembled clinical presentation of cocaine consumption with peripheral vasospasms and severe, limb-threatening ischemia. [2, 3] However, as serotonin receptor pathway agonists, NBOMes as well as LSD may evoke vasoconstriction.[4] Furthermore, the combination of the ingested substances including amphetamines (i.e. ecstasy) might have led to the distinct clinical symptoms, potentiating vasoconstriction. [5]

This limits our observation, as the patient’s clinical presentation cannot only be attributed to NBOMe ingestion. We ascribe the symptoms mostly to the substance NBOMe, as it was new in the patient’s anamnesis. Furthermore, traces of cocaine, which is also known to evoke symptoms alike [2, 3] were not found in the urinary drug screening.

3, The last paragraph focusses on schizophrenia and mental disorders. Consider to shorten as this topic is not mentioned at all in other sections of your manuscript. 

Authors´ response: Thank you for the hint, we shortened the last paragraph and included the primary disease as reported in the anamnesis.

Point-by-point response to Reviewer 2:

Thank you very much for carefully reading our manuscript and your valuable suggestions, which we have followed.

Open Review

English language and style

( ) Extensive editing of English language and style required
( ) Moderate English changes required
(x) English language and style are fine/minor spell check required
( ) I don't feel qualified to judge about the English language and style

Yes

Can be improved

Must be improved

Not applicable

Does the introduction provide sufficient background and include all relevant references?

( )

( )

(x)

( )

Is the research design appropriate?

(x)

( )

( )

( )

Are the methods adequately described?

( )

(x)

( )

( )

Are the results clearly presented?

( )

( )

(x)

( )

Are the conclusions supported by the results?

( )

( )

(x)

( )

Comments and Suggestions for Authors

The manuscript by Wadowski et al the case of a 30-year-old man admitted to the hospital due to acute ischemia of the left arm with clinical symptoms of pallor, pulselessness, paresthesia and a motoric deficit after intake of phenylethylamine derivate 25I-NBOMe. The patient had a history of schizophrenia and drug abuse and disclosed during the hospital stay the sublingual intake of a substance bought as 25I-NBOMe the night before the ischemic event. The remainder of the drugs was analyzed using high performance liquid chromatography. Drug analysis retrieved a composition of the isomers 25I- NBOMe, 25C- NBOMe, and 25H-NBOMe as well as traces of pentylon.

Interestingly, the authors mention the bio-psycho-social long-term treatment effects of 25I-NBOMe treatment on health literacy of patients.

1.The authors reported about drug abuse, in particular cocaine of the patient (1 week earlier). In addition, the authors write:

“[4] The symptoms of our patient after ingestion of NBOMe isomers resembled clinical presentation of cocaine consumption with peripheral vasospasms and severe, limb-threatening ischemia.”

Can the authors comment on possible effects of the cocaine (and or exctasy) abuse on the described symptoms?

Authors´ response: In the hospital database cocaine- abuse was documented about six weeks before the ischemic event.

The symptoms of our patient after ingestion of NBOMe isomers resembled clinical presentation of cocaine consumption with peripheral vasospasms and severe, limb-threatening ischemia. [2, 3] However, as serotonin receptor pathway agonists, NBOMes as well as LSD may evoke vasoconstriction.[4] Furthermore, the combination of the ingested substances including amphetamines (i.e. ecstasy) might have led to the distinct clinical symptoms, potentiating vasoconstriction. [5]

This limits our observation, as the patient’s clinical presentation cannot only be attributed to NBOMe ingestion. We ascribe the symptoms mostly to the substance NBOMe, as it was new in the patient’s anamnesis. Furthermore, traces of cocaine, which is also known to evoke symptoms alike [2, 3] were not found in the urinary drug screening.

This is now mentioned on page 4, line 105, page 5, lines 106-107 and page 7, lines 154-163.

It is difficult to confirm the particular effects of 25I-NBOMe as the patient admitted the intake of methylone, ethylone, lysergic acid diethylamide (LSD) and ecstasy as well as cigarette smoking. Please verify in the text.

Authors´ response: As mentioned above, we now declare the drug combination as a limitation to our observations.

What is novel in this study, as it is known about NBOMe that:

“The most frequent side effects include tachycardia, agitation and hypertension”

Authors´ response: This case report aims at raising the awareness of severe limb-threatening complications of drug abuse, in particular novel substances, which can be easily purchased via the Internet.

This is now mentioned in the discussion on page 7, lines 150-151 of the revised manuscript.

In this text passage is something wrong with references:

129 4. Discussion

130 The substance NBOMe has occurred on the illicit drug market as a new psychedelic substance.

131 [3] Effects are observed after doses of 50μg and overdosing as well as fatal poisoning is extremely

132 facile. [5] The most frequent side effects include tachycardia, agitation and hypertension. [4] The

133 symptoms of our patient after ingestion of NBOMe isomers resembled clinical presentation of cocaine

134 consumption with peripheral vasospasms and severe, limb-threatening ischemia. [9, 10] However, as

135 serotonin receptor pathway agonists, NBOMes as well as LSD may evoke vasoconstriction.[11]

136 Furthermore, our patient presented himself with hypertension and elevated creatinine kinase levels,

137 which were also described previously.[4]

138 The patient was discharged after complete resolution of the vasospasm as demonstrated by

139 duplex sonography and CTA and exclusion of a thromboembolic event. We decided to treat

140 hypertension with a calcium channel antagonist in analogy to medication in patients with (secondary)

141 Raynaud’s phenomenon. [12]

Authors´ response: Thank you for carefully reading our manuscript and the valuable comment, we have now corrected the numbering of the references.

The new paragraph reads as follows (page 7, lines 151-169 of the revised manuscript):

NBOMe isomers have occurred on the illicit drug market as a new psychedelic substance. [1] Effects are observed after doses of 50µg and overdosing as well as fatal poisoning is extremely facile. [6] The most frequent side effects include tachycardia, agitation and hypertension. [7] The symptoms of our patient after ingestion of NBOMe isomers resembled clinical presentation of cocaine consumption with peripheral vasospasms and severe, limb-threatening ischemia. [2, 3] However, as serotonin receptor pathway agonists, NBOMes as well as LSD may evoke vasoconstriction.[4] Furthermore, the combination of the ingested substances including amphetamines (i.e. ecstasy) might have led to the distinct clinical symptoms, potentiating vasoconstriction. [5]

This limits our observation, as the patient’s clinical presentation cannot only be attributed to NBOMe ingestion. We ascribe the symptoms mostly to the substance NBOMe, as it was new in the patient’s anamnesis. Furthermore, traces of cocaine, which is also known to evoke symptoms alike [2, 3] were not found in the urinary drug screening.

Other side effects observed in our patient were hypertension and elevated creatinine kinase levels, which were also described previously to be NBOMe associated.[7]

The patient was discharged after complete resolution of the vasospasm as demonstrated by duplex sonography and CTA and exclusion of a thromboembolic event. We decided to treat hypertension with a calcium channel antagonist in analogy to medication in patients with (secondary) Raynaud’s phenomenon. [8]

The are 2 references starting with number 1.

Authors´ response: Thank you for the hint, we have now restructured the references.

References

Halberstadt AL, Geyer MA. Effects of the hallucinogen 2,5-dimethoxy-4-iodophenethylamine (2C-I) and superpotent N-benzyl derivatives on the head twitch response. Neuropharmacology. 2014;77:200-7. doi:10.1016/j.neuropharm.2013.08.025. Balbir-Gurman A, Braun-Moscovici Y, Nahir AM. Cocaine-Induced raynaud's phenomenon and ischaemic finger necrosis. Clin Rheumatol. 2001;20(5):376-8. Gutierrez A, England JD, Krupski WC. Cocaine-induced peripheral vascular occlusive disease--a case report. Angiology. 1998;49(3):221-4. Kaumann AJ, Levy FO. 5-hydroxytryptamine receptors in the human cardiovascular system. Pharmacol Ther. 2006;111(3):674-706. Broadley KJ. The vascular effects of trace amines and amphetamines. Pharmacol Ther. 2010;125(3):363-75. doi:10.1016/j.pharmthera.2009.11.005. Bersani FS, Corazza O, Albano G, Valeriani G, Santacroce R, Bolzan Mariotti Posocco F et al. 25C-NBOMe: preliminary data on pharmacology, psychoactive effects, and toxicity of a new potent and dangerous hallucinogenic drug. Biomed Res Int. 2014;2014:734749. doi:10.1155/2014/734749. Suzuki J, Dekker MA, Valenti ES, Arbelo Cruz FA, Correa AM, Poklis JL et al. Toxicities associated with NBOMe ingestion-a novel class of potent hallucinogens: a review of the literature. Psychosomatics. 2015;56(2):129-39. doi:10.1016/j.psym.2014.11.002. Wigley FM, Flavahan NA. Raynaud's Phenomenon. N Engl J Med. 2016;375(6):556-65. doi:10.1056/NEJMra1507638.

Reviewer 2 Report

The manuscript by Wadowski et al the case of a 30-year-old man admitted to the hospital due to acute ischemia of the left arm with clinical symptoms of pallor, pulselessness, paresthesia and a motoric deficit after intake of phenylethylamine derivate 25I-NBOMe. The patient had a history of schizophrenia and drug abuse and disclosed during the hospital stay the sublingual intake of a substance bought as 25I-NBOMe the night before the ischemic event. The remainder of the drugs was analyzed using high performance liquid chromatography. Drug analysis retrieved a composition of the isomers 25I- NBOMe, 25C- NBOMe, and 25H-NBOMe as well as traces of pentylon.

Interestingly, the authors mention the bio-psycho-social long-term treatment effects of 25I-NBOMe treatment on health literacy of patients.

1.The authors reported about Drug abuse, in particular cocaine of the patient (1 week earlier). In addition, the authors write:

“[4] The symptoms of our patient after ingestion of NBOMe isomers resembled clinical presentation of cocaine consumption with peripheral vasospasms and severe, limb-threatening ischemia.”

Can the authors comment on possible effects of the cocaine (and or exctasy) abuse on the described symptoms?

2. It is difficult to confirm the particular effects of 25I-NBOMe as the patient admitted the intake of methylone, ethylone, lysergic acid diethylamide (LSD) and ecstasy as well as cigarette smoking. Please verify in the text.

3 What inovel in this study, as it is known about NBOMe that:

“The most frequent side effects include tachycardia, agitation and hypertension”

4. In this text passage is something wrong with references:

129 4. Discussion

130 The substance NBOMe has occurred on the illicit drug market as a new psychedelic substance.

131 [3] Effects are observed after doses of 50μg and overdosing as well as fatal poisoning is extremely

132 facile. [5] The most frequent side effects include tachycardia, agitation and hypertension. [4] The

133 symptoms of our patient after ingestion of NBOMe isomers resembled clinical presentation of cocaine

134 consumption with peripheral vasospasms and severe, limb-threatening ischemia. [9, 10] However, as

135 serotonin receptor pathway agonists, NBOMes as well as LSD may evoke vasoconstriction.[11]

136 Furthermore, our patient presented himself with hypertension and elevated creatinine kinase levels,

137 which were also described previously.[4]

138 The patient was discharged after complete resolution of the vasospasm as demonstrated by

139 duplex sonography and CTA and exclusion of a thromboembolic event. We decided to treat

140 hypertension with a calcium channel antagonist in analogy to medication in patients with (secondary)

141 Raynaud’s phenomenon. [12]

5. The are 2 References starting with number 1.

Author Response

(The authors gave the same response as above.)

Round 2

Reviewer 1 Report

This reviewer honestly appreciates the efforts made in order to prepare the revised form of the manuscript.

The concers raised by this reviewer have been adequately adressed mostly, except for some points:

1.) Section 2.1 describes routine non-invasive vascular tests, but they are not laborytory tests. Please delete „laboratory“.

2.) The abstract is still a bit confusing and not in line with the manuscript. In the main part of this manuscript, the authors mention all diagnostic features in the methods section, and findings of the diagnostic tests performed are reported in the results section. Please also use this approach for the abstract (list performed diagnostics comprising colour-coded duplex sonography, CT angiography, drug analysis in the methods section, and report findings and therapeutic steps and outcome in the results section.

3.) This reviewer recommends to add one more Paragraph to the results section:

# 3.1. routine medical and clinical diagnostics. Please consider to cut line 135-137 (The patient provided the remainder of the drug samples of NBOMe for laboratory analysis three weeks later) and paste in line 110. Amend that NBOMe sample analysis is presented in section 3.2

# 3.2 NBOMe sample analysis

# 3.3 therapeutic measures and outcome (insert line 104-105 in this paragraph, followed by 111-113. Please consider to also to cut line 133-135, 138-140 and paste to this paragraph 3.3.

4.) Please think to include line 66-67 at an appropriate position in section 1.1 (maybe line 55)

Author Response

To the Editor-in-Chief

                                                                               Vienna, 7 December 2019

Dear Professor Tchounwou,

We would like to submit a revised version of our manuscript, entitled “Acute Limb Ischemia after Intake of the Phenylethylamine Derivate NBOMe“ (Manuscript ID: ijerph-631953), for publication in the International Journal of Environmental Research and Public Health.

We would like to thank you and the referee for the review and the valuable comments and suggestions. The manuscript has been amended accordingly. We believe that these changes have resulted in a greatly improved manuscript, which we hope is now suitable for publication in International Journal of Environmental Research and Public Health.

Yours sincerely,

Patricia Wadowski, MD, PhD

Point-by-point response to Reviewer 1:

Thank you very much for carefully reading our manuscript and your valuable suggestions, which we have followed.

Open Review

(x) I would not like to sign my review report

( ) I would like to sign my review report

English language and style

( ) Extensive editing of English language and style required

(x) Moderate English changes required

( ) English language and style are fine/minor spell check required

( ) I don't feel qualified to judge about the English language and style

Yes         Can be improved     Must be improved    Not applicable

Does the introduction provide sufficient background and include all relevant references?

(x)           ( )            ( )            ( )

Is the research design appropriate?

(x)           ( )            ( )            ( )

Are the methods adequately described?

( )            (x)           ( )            ( )

Are the results clearly presented?

( )            (x)           ( )            ( )

Are the conclusions supported by the results?

(x)           ( )            ( )            ( )

Comments and Suggestions for Authors

This reviewer honestly appreciates the efforts made in order to prepare the revised form of the manuscript.

The concerns raised by this reviewer have been adequately adressed mostly, except for some points:

1.) Section 2.1 describes routine non-invasive vascular tests, but they are not laboratory tests. Please delete „laboratory“.

Authors´ response: Thank you for the suggestion, the word “laboratory” has been deleted.

2.) The abstract is still a bit confusing and not in line with the manuscript. In the main part of this manuscript, the authors mention all diagnostic features in the methods section, and findings of the diagnostic tests performed are reported in the results section. Please also use this approach for the abstract (list performed diagnostics comprising colour-coded duplex sonography, CT angiography, drug analysis in the methods section, and report findings and therapeutic steps and outcome in the results section.

Authors´ response: Thank you for the comment. We have now revised the Methods and Results in the abstract (page 1, lines 20-34).

Methods: Routine clinical diagnostics including among others color-coded duplex sonography and computed tomography angiography (CTA) were performed. The remainder of the drugs was analyzed using high performance liquid chromatography. Results: Initial color-coded duplex sonography of the upper left limb showed pathological flow profiles of the axillary, brachial, ulnar and radial artery with a reduced diameter of the ulnar (0.9mm) and radial (1.1mm) artery. In consequence, peripheral vasospasm, distal arterial thrombosis or arterial embolization was anticipated. As therapeutic measures, the patient immediately received intravenous systemic vasodilators (alprostadil) and therapeutic anticoagulation with low molecular weight heparin. Instant symptom improvement was observed within the first day after therapy initiation. The subsequently performed CTA of the heart and left arm showed no signs of thrombotic material. Treatment was continued for five days and the patient was released thereafter having completely normalized perfusion in his left arm. Outpatient treatment was continued with calcium-channel blockers, as the patient had also displayed arterial hypertension. Drug analysis retrieved a composition of the isomers 25I- NBOMe, 25C- NBOMe, and 25H-NBOMe as well as traces of pentylon.

3.) This reviewer recommends to add one more Paragraph to the results section:

# 3.1. routine medical and clinical diagnostics. Please consider to cut line 135-137 (The patient provided the remainder of the drug samples of NBOMe for laboratory analysis three weeks later) and paste in line 110. Amend that NBOMe sample analysis is presented in section 3.2

Authors´ response: As suggested, we have made the appropriate changes. This is now mentioned on page 5, lines 109-111 of the revised manuscript.

The patient provided the remainder of the drug samples of NBOMe for laboratory analyses three weeks later. NBOMe sample analysis is presented in section 3.2.

# 3.2 NBOMe sample analysis

# 3.3 therapeutic measures and outcome (insert line 104-105 in this paragraph, followed by 111-113. Please consider also to cut line 133-135, 138-140 and paste to this paragraph 3.3.

Authors´ response: As suggested, we have now included the section “3.3 Therapeutic measures and outcome” on page 6, lines 140-145 and page 7, lines 146 -151 of the revised manuscript.

3.3 Therapeutic measures and outcome

Due to the suspected acute vasospasm, arterial thrombosis (or arterial embolization) the patient received therapeutic low molecular weight heparin and intravenous alprostadil instantly. The patient´s symptoms improved constantly during the hospital stay, the motoric function restored completely within the first day of treatment initiation and the patient reported merely occasional paresthesia in the left forefinger. The patient could be discharged after five days in hospital with calcium-channel blockers as medication. One week later, a 24h blood pressure monitoring was performed in the outpatient clinic, which showed hypertension with a flattened 24h- blood pressure profile (non-dipper). In the further course, however, he did not appear to the next control visit, which was scheduled four weeks later, and it was also not possible for us to establish a telephone contact to him to evaluate the clinical outcome and to do therapeutic adjustments.

4.) Please think to include line 66-67 at an appropriate position in section 1.1 (maybe line 55)

Authors´ response: Thank you for the comment. We have now included the sentence in section 1.1 (page 2, lines 53-55 of the revised manuscript).

1.1. Case presentation:

A 30-year-old man presented with pallor and paresthesia of the left arm at the Division of Angiology of the Medical University of Vienna. Clinical examination revealed pulselessness, pain, hypaesthesia and a slight motoric deficit of the left upper extremity. The patient was suspected of acute vasospasm, arterial thrombosis or arterial embolization, and therefore hospitalized immediately. He reported sublingual ingestion of a new illicit substance purchased online. At the time of admission the patient did not remember the substance name. Moreover, he reported the intake of methylone, ethylone, lysergic acid diethylamide (LSD) and ecstasy as well as cigarette smoking. Due to schizophrenia, the patient was on quetiapine and amisulpride therapy.

Reviewer 2 Report

My comments have been addressed well

Author Response

(The authors gave the same response as above.)
